# Developments in Checkpoint Inhibitor Therapy for the Management of Deficient Mismatch Repair (dMMR) Rectal Cancer

**Alan Su** [1] , **Rodrigo Pedraza** [2] **and Hagen Kennecke** [1,3,*]

1    Providence Cancer Institute Franz Clinic, Portland, OR 97232, USA
2    The Oregon Clinic Center for Advanced Surgery, Portland, OR 97232, USA
3    Earle A Chiles Research Institute, Portland, OR 97232, USA
*    Correspondence: hagen.kennecke@providence.org

**Abstract:** Deficient mismatch repair (dMMR)/microsatellite instability-high (MSIH) colorectal cancer is resistant to conventional chemotherapy but responds to immune checkpoint inhibition (ICI). We review the standard of care in locally advanced dMMR rectal cancer with a focus on ICI. We also present a case report to highlight the treatment complexities and unique challenges of this novel treatment approach. ICI can lead to immune related adverse events (irAEs), resulting in early treatment discontinuation as well as new challenges to surveillance and surgical management. Overall, neoadjuvant ICI can lead to robust treatment responses, but its impact on durable response and organ preservation requires further study.

**Keywords:** dMMR; immunotherapy; colorectal; PD-1; CTLA-4

## 1. Introduction

Colorectal cancer is the second most common cause of cancer related death in North America [1]. Locally advanced rectal cancer is managed with trimodal therapy, which includes radiation, chemotherapy, and surgery [2–5]. However, organ preservation protocols involving close surveillance have gained traction in the management of rectal cancer [6], particularly in neoadjuvant therapies that achieve robust pathological responses. However, rectal adenocarcinomas with a deficiency in mismatch repair (dMMR), present in 2.7% of all rectal cancers [7], are resistant to usual fluoropyrimidine-based chemotherapeutic regimens [8]. Recently, immune checkpoint inhibitors (ICIs) have been shown to lead to robust responses in dMMR rectal tumors with a high clinical complete response sustained over a six-month period [9].

Previous reviews have discussed the safety, efficacy, prognostic biomarkers, nuances in testing, and response assessment of ICI in the context of metastatic colorectal cancer [10,11]. In this review article, we highlight the current standard of care in locally advanced dMMR rectal cancer with a focus on ICI. While phase II trials have shown both the favorable safety and efficacy of ICI neoadjuvant therapy, we present a case report to highlight the treatment complexities and unique challenges of this novel treatment approach.

## 2. Epidemiology and Biology

dMMR status is a hypermutable trait caused by dysfunction of the intrinsic DNA mismatch repair system [12]. dMMR is present in 15–17% of colorectal cancers and is found predominantly in right-sided malignancies [13,14]. A prior meta-analysis showed no ethnicity-specific disparities in MSI-H frequency [15]. dMMR status can be inherited as part of Lynch syndrome, leading to germline mutations in one of the mismatch repair genes that increases the risk of concomitant gastric, endometrial, ovarian, hepatobiliary, urinary, brain, and dermatologic cancers [16]. In other cases, de novo epigenetic silencing

of the MLH1 gene or other MMR genes leads to the same pathology without a heritable predisposition [17]. While up to 90% of dMMR tumors are associated with the inactivation of the MLH1 or MSH2 mismatch repair genes [18], a subset of dMMR tumors are not associated with a Lynch syndrome-associated mutation.

dMMR status confers prognostic value in that it is often associated with a reduced propensity for metastasis, and it is theorized that this is due to its nature as a source of immunogenic neo-antigens, predisposing it to intratumoral T-cell response [19]. Lynch syndrome in early-stage disease has been associated with cancer onset at an earlier age, more proximal disease, more poorly differentiated histology, mucinous subtypes, and improved overall prognosis compared with mismatch-proficient (pMMR) colorectal cancers [20–22].

### 3. Current Evidence for ICI in Non-Metastatic dMMR Colorectal Cancer

Recently, ICIs targeting programmed cell death protein 1 and its ligand (PD-1/PD-L1) have demonstrated robust responses in the treatment of both metastatic and locally advanced dMMR colorectal cancer. In 2017, the Food and Drug administration (FDA) approved nivolumab for MSI-H/dMMR metastatic colorectal cancer that has progressed following treatment with fluorpyrimidine, oxaliplatin, and irinotecan based on a 31.1% objective response rate at 12 weeks in the original CHECKMATE 142 study (NCT02060188) [23]. In 2020, a phase III trial showed that, in those with metastatic dMMR/MSI-H colorectal cancer, pembrolizumab achieved a superior response to that of fluoropyrimidine-based chemotherapy, with an overall response rate of 43.8% vs. 33.1%, ongoing responses of 83% vs. 35% at 24 months, and progression-free survival of 16.5 vs. 8.2 months [24]. The addition of CTLA-4 blockade increases the efficacy of PD-1 therapy. In 2018, the FDA granted accelerated approval to 1 mg/kg ipilimumab for use in combination with 3 mg/kg nivolumab for those with dMMR metastatic colorectal cancer based on the CheckMate142 study. In this phase II multi-center trial of 119 patients, combination nivolumab–ipilimumab led to greater progression-free survival and overall survival compared with monotherapy with nivolumab alone [25].

With promising results in the treatment of metastatic disease, ICI has been applied to early-stage colorectal cancer. Of importance in this context is that earlier stages of colorectal cancer have a higher prevalence of microsatellite instability (20% in stages I and II, 12% in stage III) compared with when it has metastasized (4–5%) [13,26], suggesting a wider applicability of immunotherapy in this tumor stage. A case-series by Zhang showcased 2 patients with stage III dMMR rectal cancer who were treated with nivolumab (3 mg/kg) every 2 weeks for 6 cycles followed by total mesorectal excision; both patients achieved clinical complete response while one patient was confirmed to have pathological complete response [27]. The VOLTAGE-A study investigated nivolumab monotherapy in 42 patients with early-stage (II–III) colorectal cancer (5 of whom had dMMR colorectal cancer) and achieved a 33% pathological complete response (pCR), with dMMR tumors showing robust responses (60%) [28]. The NICHE-II study (NCT030264140) investigated combination nivolumab and CTLA-4 inhibitor ipiliumumab therapy in 41 patients with early-stage (I–III) colorectal adenocarcinoma [29]. A pCR was observed in 20/20 patients (100%) with dMMR tumors, and major pathologic response (MPR) was achieved in 19/20 (95%) of the dMMR tumors. In pMMR tumors, patients in the VOLTAGE-A and NICHE-II studies experienced less robust MPRs (30% and 20%, respectively). Recent studies of ICIs in early-stage colorectal cancer are summarized in Table 1.

**Table 1.** Recent Studies of Immune Checkpoint Inhibitors for the Treatment of Early-Stage Colorectal Cancer.

| Author | Study | Year | dMMR/ pMMR | Location | Stage | Neoadjuvant Therapy Strategy | Efficacy | Adverse Events |
|---|---|---|---|---|---|---|---|---|
| Zhang [27] | Case Series N = 2 | 2019 | dMMR: 2 | Rectal | III | Both patients underwent nivolumab 3 mg/kg every 2 weeks for 6 cycles followed by total mesorectal excision. | 1. pCR 2. cCR | No Grade 3–4 adverse events (0%). |
| Bando [28] | VOLTAGE-A (Phase II) N = 44 | 2020 | dMMR: 5 pMMR: 39 | Rectal | II–III | Nivolumab 240 mg every 2 weeks ×5 cycles followed by radical surgery. | pCR dMMR 3/5 (60%) pMMR 11/37 (30%) | Grade 3 3/42 (4%) |
| Chalabi [29] | NICHE (Phase II) N = 41 | 2020 | dMMR: 21 pMMR: 20 | Colon | I–III | Ipilimumab (1 mg/kg) on day 1 followed by nivolumab (3 mg/kg) on day 1 + 15. Patients with pMMR tumors randomly assigned to receive celecoxib from day 1 until the day before surgery in addition to immunotherapy. Surgery was performed within 6 weeks following the last day of neoadjuvant therapy. | pCR dMMR 20/20 (100%) pMMR 4/15 (27%) MPR dMMR 19/20 (95%) pMMR 3/15 (20%) | Grade 3 5/40 (12%) |
| Demisse [30] | Case Series N = 3 | 2020 | dMMR: 3 | Rectal | II–III | 1. Pembrolizumab 200 mg every 3 weeks for 11 cycles. 2. Nivolumab 3 mg/kg and ipilimumab 1 mg/kg intravenously every 3 weeks for 7 cycles. 3. FOLFOX with concurrent pembrolizumab for 7 cycles followed by low anterior resection. | pCR 3/3 (100%) | 1 patient discontinued treatment due to grade II fatigue. No grade 3–4 adverse events were reported. |
| Liu * [31] | Case Series N = 8 | 2020 | dMMR: 8 | Colorectal | II–IV II–III: 4 IV: 4 | 1. Pembrolizumab 240 mg for 2 cycles with neoadjuvant XELOX (oxaliplatin and capecitabine) followed by subtotal colectomy. 2. Pembrolizumab 200 mg + ipilimumab 50 mg for 4 cycles. 3. Nivolumab 140 mg for 12 cycles and anterior resection. 4. Pembrolizumab 200 mg for 4 cycles followed by right hemicolectomy with lymph node dissection. | 1. pCR 2. No response 3. pCR 4. pR | Grade 3 1/8 (13%) |
| Avallone [32] | NICOLE (Phase II) N = 22 | 2021 | dMMR: 3 pMMR: 19 | Colon | I–III | Nivolumab 240 mg on day 1 + 15. Surgery after 5 weeks. | MPR dMMR 0/3 (0%) pMMR 3/19 (16%) | Grade 3 1/22 (5%) |
| Lin [33] | NCT04231552 (Phase II) N = 29 | 2021 | dMMR: 1 pMMR: 26 | Rectal | II–III | 5 × 5 Gy short course radiation therapy followed by 21 days of CAPOX (oxaliplatin 130 mg/m$^2$ intravenously, day 1; capecitabine 1000 mg/m$^2$ orally twice daily, days 1–14) plus camrelizumab (200 mg intravenously, day 1), followed by radical surgery after 1 week. | pCR dMMR 1/1 (100%) pMMR 12/26 (46%) | Grade 3 8/30 (26.7%) |
| Salvatore [34] | AVANA (Phase II) N = 101 | 2021 | dMMR: 1 pMMR: 38 Other: 57 | Rectal | II–III | CTRT (capecitabine 825 mg/sqm/bid 5 days/week + 50.4 Gy in 28 fractions over 5.5 weeks) pluse 6 cycles of avelumab 10 mg/kg every 2 weeks followed by total mesorectal excision at 8–10 weeks after the end of CTRT. | pCR Total 22/96 (23%) MPR Total 59/96 (62%) | Grade 3 8/96 (8%) Grade 4 4/96 (4%) Avelumab interrupted in 9/101 (9%) due to treatment toxicity. |

**Table 1.** *Cont.*

| Author | Study | Year | dMMR/ pMMR | Location | Stage | Neoadjuvant Therapy Strategy | Efficacy | Adverse Events |
|---|---|---|---|---|---|---|---|---|
| Cercek [9] | NCT04165772 (Phase II) N = 12 | 2022 | dMMR: 12 | Rectal | II–III | Dostarlimab (500 mg) administered every 3 weeks for 6 months followed by standard radiation therapy (5040 cGy in 28 fractions) with concurrent administration of capecitabine. Patients with clinical complete response after induction of anti-PD1 or chemoradiotherapy underwent non-operative follow-up. | cCR dMMR 12/12 (100%) | No Grade 3–4 adverse events (0%). |
| Hu [35] | PICC (Phase II) N = 53 | 2022 | dMMR: 34 | Colorectal | II–III | 34 participants randomly assigned to either toripalimab (n = 17) or toripalimab + celecoxib (n = 17) for 6 months followed by colectomy. Toripalimab was administered every 2 weeks for 6 months. Celecoxib group received additional 200 mg oral celecoxib twice daily for 6 months. | Toripalimab pCR 11/17 (65%) Toripalimab + celecoxib pCR 15/17 (88%) | Grade 3 1/34 (3%) |
| Ludford [36] | NCT04082572 (Phase II) N = 35 | 2023 | dMMR: 27 | Colorectal | II/II | Pembrolizumab 200 mg once every 3 weeks for 6 months followed by surgical resection with an option to continue therapy for 1 year followed by observation. | pCR 11/14 (79%) | Grade 3 2/35 (6%) |

N = total number of participants enrolled; MPR = major pathological response (<10% viable tumor cells); pCR = pathologic complete response; cCR = clinical complete response; pR = partial response; dMMR = mismatch repair deficient/microsatellite instability-high; pMMR = mismatch repair proficient/microsatellite stable. * This was a case series of 8 patients split into metastatic (stage IV) and locally advanced (stage III) MSI-H tumors. Only the tumor responses of locally advanced tumors were reported here.

Though durable responses have been observed, a subset of locally advanced dMMR/MSI-H colorectal tumors do not achieve clinical response with ICI. In early-stage colorectal cancer, PD-1/PD-L1 expression status is associated with positive prognostic markers for overall survival and disease-free survival [37], but the level of PD-1/PD-L1 expression as a marker of responsiveness to ICI remains to be studied. Studies of tumor mutational burden (TMB) and high-MSI status suggest that states of high neoantigen expression are more prone to lymphocyte infiltration, particularly with host-mediated anti-tumoral responses that are enhanced with ICI [38,39]. A meta-analysis of 43 trials involving 21,015 metastatic colorectal cancer patients showed that tumors with high tumor-infiltrating lymphocyte (TIL) scores were associated with high overall survival, cancer-specific survival, and disease-free survival [40]. Of the lymphocyte subtypes, CD8+ cytotoxic T-cell infiltrates showed the highest association with patient survival [41]. Other extrinsic inflammatory biomarkers associated with longer progression free survival include a high T-cell-inflamed gene expression profile (GEP), a low T-cell dysfunction and exclusion gene signature (TIDE), a low melanocytic plasticity signature (MPS), and high B-cell-focused gene signature expression [42]. One of the limitations of these studies is that they included patients with dMMR and pMMR tumors with varying tumor stages; positive prognostic biomarkers for ICI-responsiveness specific to locally advanced dMMR colorectal tumors have yet to be established.

## 4. Current Management of dMMR Rectal Cancer

Current management of both preserved pMMR and dMMR rectal cancers includes preoperative chemoradiation and perioperative fluoropyrimidine-based chemotherapy [5]. A large retrospective series of stage I–IV dMMR rectal tumors showed a clinical completer response (cCR) of 28% with chemoradiation and rectal cancer specific survivals of 85% and 60% in stage II–III and IV tumors, respectively [43]. For patients with metastatic or unresectable dMMR rectal cancer, pembrolizumab or nivolumab, as a monotherapy or in combination with ipilumumab or FOLFOX chemotherapy, are recommended, with some reports documenting a high rate of response with ICI alone.

For patients with stage II/III tumors, monotherapy with the anti-PD1 agent dostarlimab in a small single-arm phase II study achieved a clinical response rate of 100% (95% CI, 74 to 100), with no evidence of a tumor on magnetic resonance imaging, endoscopic evaluation, digital rectal examination, or biopsy [8]. No patients received chemoradiation or underwent mesorectal excision, and no adverse events of grade 3 or higher were reported. Though promising, a larger multicenter trial with a longer follow-up is needed.

Other clinical trials currently in progress are synergizing ICI with radiotherapy in locally advanced colorectal tumors. As is described in Table 1, varying radiation doses have been paired with ICI in the treatment of dMMR/MSI-H colorectal cancers, e.g., with capecitabine/oxaliplatin/camrelizumab, capecitabine/avelumab, and dostarlimab. Other trials in progress involve pembrolizumab (NCT04109755), sintilimab (NCT04304209), toripalimab (NCT04301557), and durvalumab (NCT03102047). External beam radiation therapy may generate neo-antigens and lead to an immune-based anti-tumor response outside the target area in what is known as the abscopal effect [44,45]. Two case reports of metastatic colon cancer treated with carbon-ion radiation therapy at 73.6 Gy over 16 fractions or 50.4 Gy over 12 fractions decreased the tumor size in both the targeted area as well as the non-targeted area, with durable response [46]. NRG-GI002 is a phase II clinical trial with 185 stage II/III MSS rectal cancer patients that pairs pembrolizumab to a FOLFOX/capecitabine/50.4 Gy chemotherapy regimen followed by mesorectal excision at 8–12 weeks [47]. The addition of pembrolizumab as part of the chemoradiotherapy regimen improved overall survival at 3 years, but failed to improve the neoadjuvant rectal cancer (NAR) score and disease-free survival. Whether dMMR/MSI-H status affects the responsiveness of ICI to a similar chemoradiotherapy regimen remains unclear. Further studies are warranted to establish which subsets of locally advanced CRC patients would most benefit from neoadjuvant chemoradiation, particularly if a pathologic response can be achieved in the absence of radiation.

## 5. Case Report: ICI with Total Remission in dMMR laRC after a Single Cycle

A 61-year-old female patient initially presented (January 2022) with 5 months of tenesmus, increased frequency of stools, and rectal bleeding. The initial suspicion of malignancy was confirmed with contrast-enhanced tomography (Figure 1A), magnetic resonance imaging (Figure 2A), and endoscopy (Figure 3A). A fungating mass was biopsied on endoscopy and found to be a moderately differentiated invasive adenocarcinoma. Immunohistochemical stains revealed intact MLH1, MSH2, and MSH6, but absent PMS2. The tumor stage was clinical T3dN2M0 (AJCC 8th edition). There was insufficient tumor tissue available for next generation sequencing. Subsequent germline testing revealed the absence of germline defect in mismatch repair.

The patient commenced therapy on a clinical trial (February 2022) with 480 mg nivolumab and 3 mg/kg ipilumumab. Eleven days later, she presented with diarrhea, nausea, skin rash, elevated transaminases and autoimmune hepatitis, and evidence of adrenal insufficiency, and she required two months of steroid therapy with prolonged taper starting at 80 mg prednisone twice daily. Due to the severity of the immune-related events, the patient was not eligible for further ICI therapy on the clinical trial, and treatment was discontinued and a decision was made after the first cycle to proceed with tumor restaging.

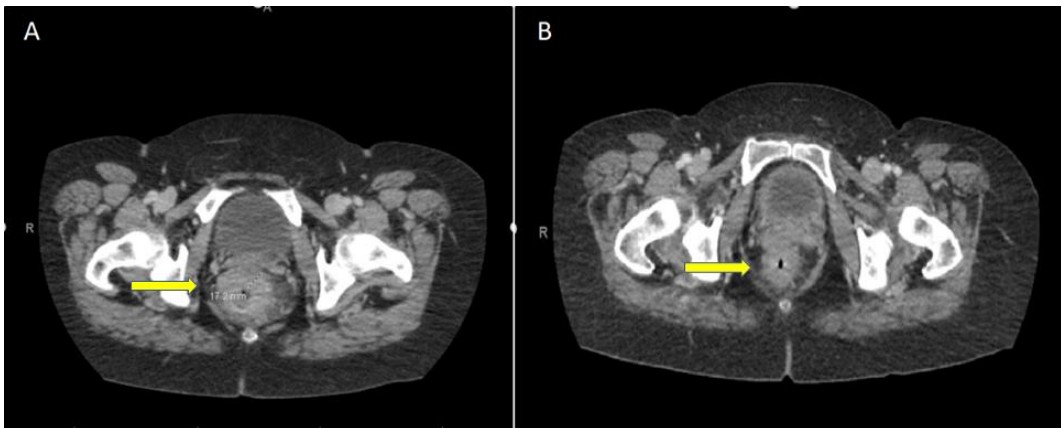

**Figure 1.** Computed tomography (CT) imaging of the rectum. (**A**) Initial CT shows irregular wall thickening of the rectum with eccentric segments of wall thickening measuring up to 1.7 cm spanning 6 cm above the rectum to the anal verge, with hazy infiltration in the perirectal fat. (**B**) Follow-up CT at 12 weeks post-immunotherapy shows no residual obstructing mass.

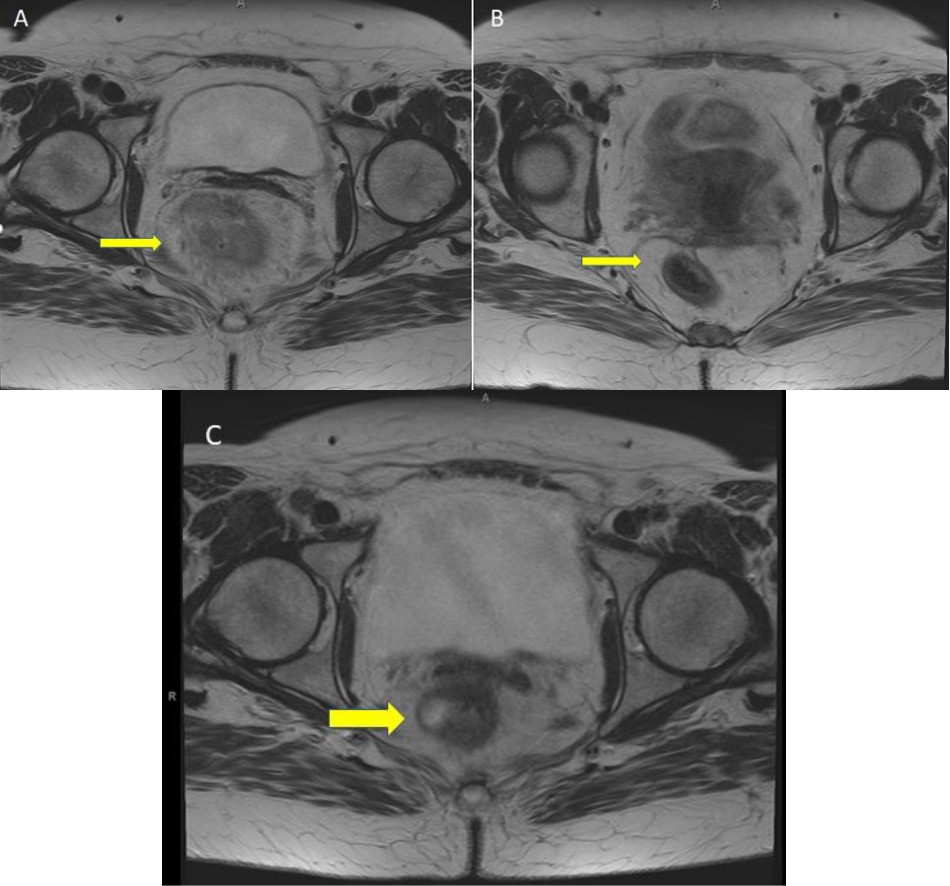

**Figure 2.** Magnetic resonance imaging (MRI) of the rectum. (**A**) T2 imaging shows a rectal mass 7.1 cm in length, 4.7 cm in circumferential depth, and a 2.2 cm extramural invasion in the perirectal fat (T3d) with irregularly bordered mesorectal lymph node (N2) and no evidence of metastases. (**B**) Follow-up T2 imaging at 12 weeks post-immunotherapy shows a 2.7 cm area of increased T2 sign and thickening of the distal rectum consistent with stricturing that was negative for malignancy of pathology. (**C**) Follow-up T2 imaging at 1 year post-immunotherapy shows an asymmetric lower rectal thickening that is 1.2 cm thick located at the previous stricture site.

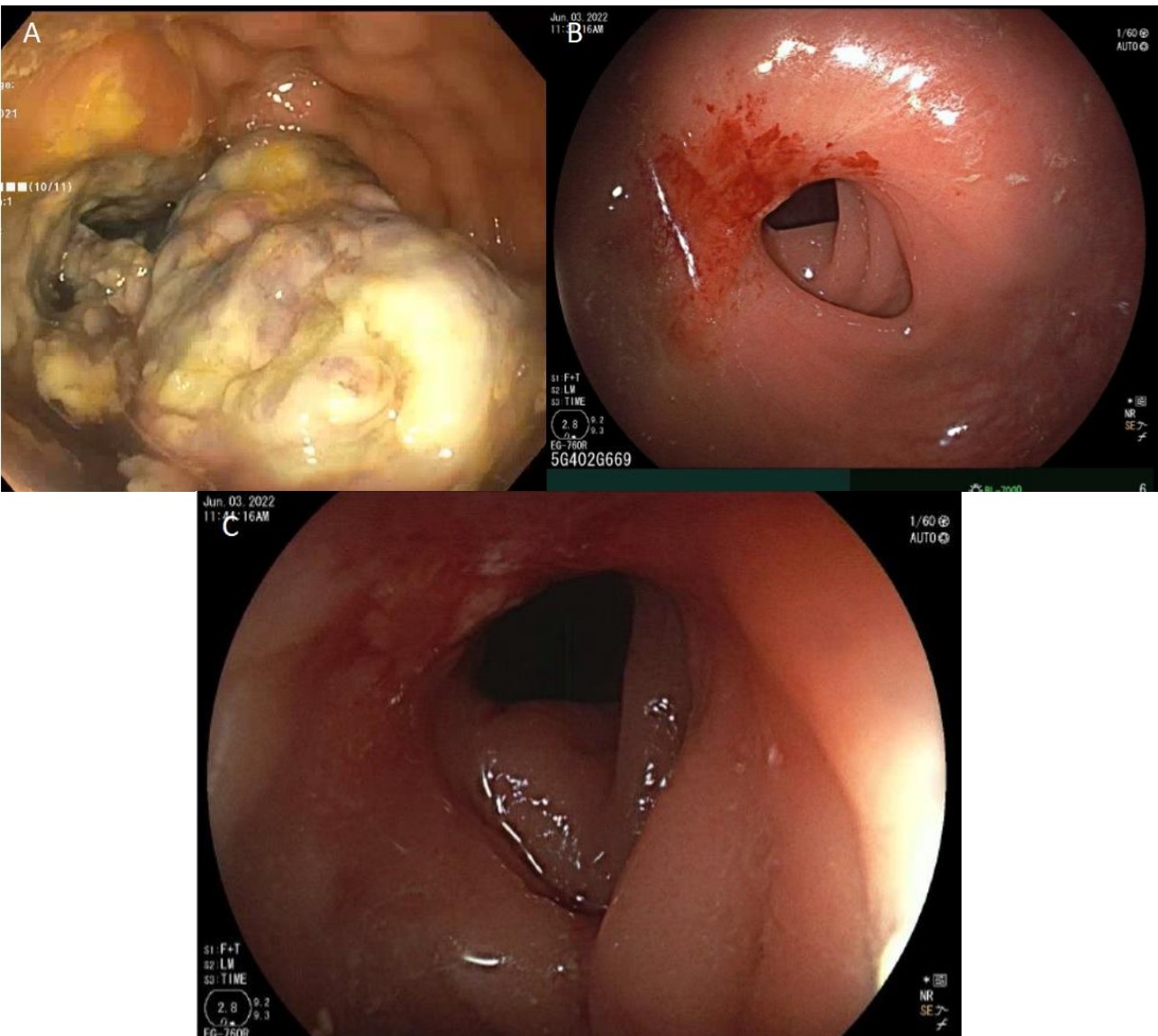

**Figure 3.** Endoscopic evaluation of rectum. (**A**) Pre-treatment view of original fungating mass observed in the rectum consistent with locally advanced dMMR rectal cancer. (**B**) Follow-up 12 weeks post-treatment shows evidence of symptomatic stricturing 8cm from the anal verge with no evidence of malignancy of pathology. (**C**) The 1 cm (length) × 1.2 cm (inner diameter) stricture was dilated to 1.5 cm with an endoscopic balloon.

At 12 weeks (May 2022) post-day 1 of treatment, with only one dose of nivolumab and ipilumumab, there was no radiographic evidence of malignancy per CT (see Figure 1B) and MRI (see Figure 2B); restaging flexible sigmoidoscopy showed no evidence of active malignancy in the tumor bed (see Figure 3B). A biopsy of the rectum showed glandular reactive change and lymphoid aggregates, but no evidence of dysplasia or malignancy, suggesting complete pathologic response. Endoscopy further revealed a benign-appearing intrinsic mild stricture measuring 1 cm in length and 1.2 cm in diameter 8 cm above the anal verge, proximal to the location of the initial malignancy (Figure 3B). Given that the patient was symptomatic from the aforementioned stricture, dilation with a 12–13.5–15 mm colonic balloon was subsequently performed under fluroscopic guidance, with improvement in luminal narrowing to 15 mm without evidence of perforation (Figure 3C). Due to ongoing symptoms of abdominal pain, tenesmus, and incomplete evacuation at 28 weeks

(19 September 2022) post-treatment, the patient underwent a second endoscopic dilation with symptomatic improvement.

At one year post-treatment, there was no evidence of tumor reoccurrence or metachronous spread of disease according to MRI (31 January 2023, Figure 2C) and endoscopy (1 February 2023). A biopsy of the previous stricture site showed normal squamo-glandular junctional mucosa with crypt architectural disarray and reactive epithelial changes without evidence of adenocarcinoma. The overall findings suggest a durable response after a single dose of combination nivolumab/ipilimumab at one year follow-up.

## 6. ICI Toxicity Management

Despite promising treatment responses, immunotherapy can lead to significant adverse side effects. Moreover, irAEs may be steroid-refractory, necessitating the use of biologics, surgical management, and early treatment discontinuation [48–51]. In monotherapy, immune checkpoint inhibitors typically have grade 3 or higher adverse events at an incidence less than 10%. In contrast, immune-related adverse events (irAE) are more common with combination immunotherapy. In the CheckMate-142 study of ipilimumab and nivolumab, Grade 3 and 4 irAEs were observed in 32% and 13% of patients, respectively, and these led to treatment discontinuation. In a random-model meta-analysis representing 4677 patients that received nivolumab–ipilimumab regimens [52], an estimated 40.6% of patients (95% CI 35.7–45.5) experienced a grade 3 or higher adverse event, and an estimated 28.3% of patients (95% CI 23.7–32.8) discontinued their treatment due to adverse events. Common adverse events include fatigue (27.9%, CI 22.6–33.3), pruritus (24.6%, CI 20.3–28.8), rash (24.0%, CI 19.3–28.7), elevated aspartate aminotransferase (21.2%, CI 14.9–27.5), and elevated alanine aminotransferase (18.1%, CI 13.1–23.2). In reference to the adrenal insufficiency in our case, endocrine irAEs include hypothyroidism (13.1%, CI 11.2–15.1), hyperthyroidism (11.0%, CI 7.7–14.4), adrenal insufficiency (4.8%, CI 2.8–6.7), and hypopituitarism (9.5%, CI 5.7–13.2). GI manifestations following combination immunotherapy include diarrhea (26.0%, CI 21.5–30.5), nausea (15.1%, CI 12.1–18.1), decreased appetite (12.1%, CI 10.3–14.0), vomiting (8.6%, CI 5.9–11.4), and colitis (8.2%, CI 5.5–10.8). Rectal stricturing was not a previously documented side effect of immunotherapy, combination, or monotherapy, and further studies are required to understand the true incidence of this rare complication.

## 7. Response Assessment and Management of Immunotherapy-Related Rectal Stricture

Standardized criteria have been developed to define tumor responses after neo-adjuvant therapy based on MRI imaging, endoscopic assessment, and digital rectal examination. Clinical responses are classified as complete, near complete, and incomplete responses, and patients experiencing a complete or near complete response may be considered for a watch-and-wait approach [6], while those with incomplete responses would be advised to proceed with definitive surgical resection.

The presence of a rectal stricture in the tumor bed presents a special diagnostic challenge to surgeons because this may be interpreted either as a palpable tumor nodule or only a partially responsive disease. While rectal stricturing has been reported following neoadjuvant chemoradiotherapy, the frequency and severity of this complication after ICI are still unknown. While a rectal stricture may represent residual tumor scar tissue, its formation may be increased by ICI-induced inflammation and colitis in and around the tumor bed. Similarly, strictures have been reported in patients diagnosed with inflammatory bowel disease (IBD) [53,54], which can be associated with an increased risk of severe GI adverse events [55,56] when treated with immune checkpoint inhibitors. The patient in our cases study had no prior history of IBD and in follow-up had no further symptoms of colitis.

The decision of whether to dilate a symptomatic rectal stricture needs to be weighed with the risk of perforation [57]. If strictures cannot be dilated, sometimes surgical resection

may be required, but this is undesirable, especially in distally located tumors where resection affects sphincteric function.

## 8. Conclusions

While previously reserved for use in metastatic disease, there is growing interest in ICI inhibitors for use in the treatment of locally advanced dMMR rectal cancers. While standard management continues to include pre-operative chemoradiation, eligible patients with dMMR rectal tumor should be offered participation in immunotherapy trials.

The development of a tumor-related stricture can present a unique challenge to the treatment and surveillance of locally advanced rectal cancer. Larger prospective studies with longer follow-ups are necessary to determine which patients would most benefit from combination or mono-immunotherapy, the role of whole pelvic irradiation, and the risk stratifying who would be most appropriate for a watch-and-wait approach based on the robustness of the tumor response.

**Author Contributions:** Conceptualization, A.S. and H.K.; abstract, A.S. and H.K.; literature search and acquisition, A.S. and H.K.; drafting and revising the manuscript, A.S., R.P. and H.K.; critical revision of the manuscript for intellectual content, A.S. and H.K. All authors have read and agreed to the published version of the manuscript.

**Funding:** This study received no external funding.

**Acknowledgments:** The assistance provided by Pedraza, who provided surgical expertise, is greatly appreciated.

**Conflicts of Interest:** The authors declare no conflict of interest.

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
