# Peer review of "Developments in Checkpoint Inhibitor Therapy for the Management of Deficient Mismatch Repair (dMMR) Rectal Cancer"

_curroncol, doi:10.3390/curroncol30040279_

Round 1

Reviewer 1 Report

Well written review article

Author Response

Thank you very much for your response to our article.

Reviewer 2 Report

In this review article, the authors aimed to highlight the current standard of care in deficiency in mismatch repair (dMMR) locally advanced rectal cancer with a focus on immune checkpoint inhibitors (ICI) as they also present a case report to highlight the treatment complexities and unique challenges of this treatment approach. They concluded that neoadjuvant ICI can lead to robust treatment responses and stated that the impact of this therapy on durable response as well as organ-preservation protocols require further study. The manuscript throws a light upon a novel treatment method, includes a relevant report of an interesting case and is adequately written. I congratulate the authors for their successful work.   

Author Response

Thank you very much for your careful review of our article and we appreciate your contribution.

Reviewer 3 Report

The authors conducted an evidence-based review of the developmental therapeutics of checkpoint inhibitors in the management of mismatch repair deficient rectal cancer.

Comments:

1. In the abstract section, change “Mismatch repair-deficient (dMMR)” to “deficient Mismatch repair (dMMR)”

2. In table 1,

A. the number of studied patients should be added to the table.

B. In the study of Cercek et al, all patients were treated with dostarlimab alone and due to the achievement of a pathologic complete response in all patients, no patients received chemoradiation or underwent surgery.

3. To improve the manuscript, adding the following issues are suggested:

A. In the section "Response assessment and management of immunotherapy”, provide potential and known biomarkers predicting response to Immunotherapy in dMMR Colorectal Cancers.

B. In the discussion section, provide some evidence regarding the synergistic effect of radiotherapy and immunotherapy; as well as, the Abscopal effect.

4. The manuscript should be checked for grammar mistakes and typos in the text and the tables.

Author Response

  1. "In the abstract section, change “Mismatch repair-deficient (dMMR)” to “deficient Mismatch repair (dMMR)”As requested, we have changed the title to deficient mismatch repair.
  2. "In table 1, the number of studied patients should be added to the table. " As requested, we provided the total number of studied patients to Table 1. Please see attachment.
  3. "In the study of Cercek et al, all patients were treated with dostarlimab alone and due to the achievement of a pathologic complete response in all patients, no patients received chemoradiation or underwent surgery." We have also clarified this in the text and in the table. Please see attachment.
  4. "In the section "Response assessment and management of immunotherapy”, provide potential and known biomarkers predicting response to Immunotherapy in dMMR Colorectal Cancers." We have included the section related to biomarkers under "Current evidence for ICI in non-metastatic dMMR colorectal cancer". Please see attachment.
  5. "In the discussion section, provide some evidence regarding the synergistic effect of radiotherapy and immunotherapy; as well as, the Abscopal effect." In the section titled “Current management of dMMR rectal cancer” we have included information about radiotherapy and the abscopal effect. 

  1. "The manuscript should be checked for grammar mistakes and typos in the text and the tables. " We have made numerous edits to fix grammatical errors and typos. 

Thank you so much for your insightful feedback of our review.

Reviewer 4 Report

Authors of the article presented a case report, which was packaged in a review article. There are serious issues with the article.

1. Most of the review part is very superficial and the case report also needs further details.

2. It has to be highlighted that there are much more detailed articles about the same topic, including in another MDPI journal. E.g., PMID 36291761 (Cancers, 14(20): 4974) and 35351582, and https://ascopubs.org/doi/full/10.1200/EDBK_349557.

3. My recommendation for the authors is to completley remove the superficial review parts, and authors should focus on the case presentation as Current Oncology allows the publication of Case Reports. However, the case reporting must be more detailed including when are where the patient was treated, details about treatment outcome is also a must. Discussion and literature data must focus only on those which are related to the details and take-home-message of the case presentation.

4. Moderate English corrections are needed.

Author Response

1) Most of the review part is very superficial and the case report also needs further details.

Your feedback is most appreciated. We provided a comprehensive review of prior published literature on ICI in the treatment of locally advanced dMMR colorectal cancer. Our review focuses specifically on the neoadjuvant setting for early-stage disease which differs from previous reviews that focus on metastatic colorectal cancer.

2) It has to be highlighted that there are much more detailed articles about the same topic, including in another MDPI journal. E.g., PMID 36291761 (Cancers, 14(20): 4974) and 35351582, and https://ascopubs.org/doi/full/10.1200/EDBK_349557.

Thank you for pointing out the articles PMID36291761 and 35351582. These provide a comprehensive overview of ICI in metastatic dMMR colorectal cancer. We decided to include these review articles as follows, "Previous reviews discussed the safety, efficacy, prognostic biomarkers, nuances in testing, and response assessment of ICI in the setting of metastatic colorectal cancer [10-11]. In this review article, we highlight the current standard of care in locally advanced dMMR rectal cancer with a focus on ICI. "

3) My recommendation for the authors is to completley remove the superficial review parts, and authors should focus on the case presentation as Current Oncology allows the publication of Case Reports. However, the case reporting must be more detailed including when are where the patient was treated, details about treatment outcome is also a must. Discussion and literature data must focus only on those which are related to the details and take-home-message of the case presentation.

For the case report, we defined our treatment outcomes as radiographic, endoscopic, and pathologic evidence of tumor regression to suggest a clinical complete response. We added information about the age of the patient and timeline of treatment. In addition, we’ve included additional radiographic/pathologic/endoscopic evaluation for 1 year post-treatment to support our findings of durable response. We did not include the name of the hospital for the sake of patient confidentiality.

4) Moderate English corrections are needed.

We have checked for grammatical errors to the best of our ability.

Reviewer 5 Report

I have no additional questions for the authors or comments on the article.

Author Response

We greatly appreciate your review and critique of our article. We have addressed grammatical errors and typos to the best of our ability.

Round 2

Reviewer 3 Report

Dear Authors,

Thank you for your revision.

Reviewer 4 Report

The article and its focus improved significantly. It has to be highlighted, however, that the novelty of the paper is minimal still.

Line 229: "Carcinoembryonic antigen (CEA) (06/08/2022) and. Due to". The sentence was not finished, please correct.